# Peptide Vaccination against Cytomegalovirus Induces Specific T Cell Response in Responses in CMV Seronegative End-Stage Renal Disease Patients

**DOI:** 10.3390/vaccines9020133

**Published:** 2021-02-06

**Authors:** Claudia Sommerer, Anita Schmitt, Angela Hückelhoven-Krauss, Thomas Giese, Thomas Bruckner, Lei Wang, Paul Schnitzler, Stefan Meuer, Martin Zeier, Michael Schmitt

**Affiliations:** 1Department of Nephrology, University Hospital Heidelberg, University of Heidelberg, 69117 Heidelberg, Germany; martin.zeier@med.uni-heidelberg.de; 2German Center for Infection Research DZIF, 69117 Heidelberg, Germany; giese@uni-hd.de (T.G.); stefan.meuer@med.uni-heidelberg.de (S.M.); 3Department of Internal Medicine V, University of Heidelberg, 69117 Heidelberg, Germany; anita.schmitt@med.uni-heidelberg.de (A.S.); angela.hueckelhoven-krauss@med.uni-heidelberg.de (A.H.-K.); Lei.wang@med.uni-heidelberg.de (L.W.); michael.schmitt@med.uni-heidelberg.de (M.S.); 4Institute of Immunology, University of Heidelberg, 69117 Heidelberg, Germany; 5Institute of Medical Biometry and Informatics, University of Heidelberg, 69117 Heidelberg, Germany; bruckner@imbi.uni-heidelberg.de; 6Department of Virology, University Hospital Heidelberg, University of Heidelberg, 69117 Heidelberg, Germany; paul.schnitzler@med.uni-heidelberg.de

**Keywords:** cytomegalovirus (CMV), CMV reactivation, phosphoprotein 65 (pp65) peptide vaccination, specific T cells, renal transplantation

## Abstract

Introduction: Cytomegalovirus (CMV) reactivation occurs in seronegative patients after solid organ transplantation (SOT) particularly from seropositive donors and can be lethal. Generation of CMV-specific T cells helps to prevent CMV reactivation. Therefore, we initiated a clinical phase I CMVpp65 peptide vaccination trial for seronegative end-stage renal disease patients waiting for kidney transplantation. Methods: The highly immunogenic nonamer peptide NLVPMVATV derived from CMV phosphoprotein 65(CMVpp65) in a water-in-oil emulsion (Montanide™) plus imiquimod (Aldara™) as an adjuvant was administered subcutaneously four times biweekly. Clinical course as well as immunological responses were monitored using IFN-γ ELISpot assays and flow cytometry for CMV-specific CD8^+^ T cells. Results: Peptide vaccination was well tolerated, and no drug-related serious adverse events were detected except for Grade I–II local skin reactions. Five of the 10 patients (50%) mounted any immune response (responders) and 40% of the patients presented CMV-specific CD8^+^ T cell responses elicited by these prophylactic vaccinations. No responders experienced CMV reactivation in the 18 months post-transplantation, while all non-responders reactivated. Conclusion: CMVpp65 peptide vaccination was safe, well tolerated, and clinically encouraging in seronegative end-stage renal disease patients waiting for kidney transplantation. Further studies with larger patient cohorts are planned.

## 1. Introduction

Renal allograft recipients are at high risk for cytomegalovirus (CMV) infection, particularly during the first three months after transplantation due to high initial immunosuppression [1]. Ten to fifty percent of renal allograft recipients develop symptomatic CMV disease. Transmission of CMV infection occurs by endogenous reactivation, by donor-derived infection transmitted via the allograft, or by de novo infection. The range of clinical manifestations of CMV infection is very broad: CMV infection may occur as asymptomatic viremia or lead to more general symptoms like fever and bone marrow suppression (CMV syndrome). CMV viremia may develop into symptomatic CMV disease with tissue invasion, which increases morbidity and mortality in transplant recipients. CMV infection is recognized as a risk factor for other poor short-term outcomes including acute allograft rejection. Not only CMV disease but also subclinical CMV infection correlates with increased long-term morbidity, graft loss [2,3], diabetes [3,4], atherosclerosis [5,6,7], neoplasia [8], and mortality [2,3].

Antiviral prophylaxis is standard of care at least in patients with CMV high-risk constellation, i.e., donor CMV-seropositive/recipient seronegative (D+R−). Its efficacy has been demonstrated in large randomized multicenter trials [9,10]. Antiviral CMV prophylaxis reduces the incidence of CMV disease and other infectious complications (e.g., other herpes, polyoma, and noroviruses). Prophylactic treatment of CMV is often associated with pronounced side effects as hematological toxicity requiring reduction of immunosuppression. Late CMV infection might occur after cessation of CMV prophylaxis three or six months after transplantation [11].

No commercial CMV vaccine is currently available. Several products are under investigation in phase I–III clinical trials: attenuated viruses, truncated proteins, and DNA vaccines [12,13,14,15,16,17]. Cellular immune response is essential for controlling CMV infection [18]. Patients might be protected, once a detectable T cell response against CMV has been reached. Recently, a first phase I trial in patients after hematopoietic stem cell transplantation has shown that this CMV peptide vaccination was safe, well tolerated, and efficacious [19].

The aim of the present trial was to prove safety and feasibility of a CMVpp65-derived vaccine in CMV-seronegative end-stage renal disease (ESRD) patients on the kidney transplant waiting list.

## 2. Material and Methods

### 2.1. Study Design

The CMVPepVac study (RCHD-CMV-1001, EudraCT No. 2012-002486-35; ISRCTN11842403) was a prospective phase I trial in CMV negative end-stage renal disease patients prior to renal transplantation performed at the Renal Clinic Heidelberg and the University Hospital Heidelberg, Germany. The study protocol was approved by the ethical committee (IRB No. AFmo-256/2013) as well as by Federal Regulatory Authority, the Paul-Ehrlich-Institute, Langen, Germany (PEI registration No. 1855/02).

The primary objective of this phase I clinical trial was to test the safety and feasibility of CMV peptide vaccination. Secondary objectives were the evaluation of cellular and humoral immune responses to the virus and the assessment of the CMV antigenemia status before and after peptide vaccination.

Main inclusion criteria were age ≥ 18 years, end-stage renal disease patients, CMV IgG seronegative, HLA-A*02 expression positivity, liver function tests (alanine aminotransferase, aspartate aminotransferase, alkaline phosphatase, and gamma-glutamyl-transpeptidase) below the threefold of the normal upper values (ultraviolet test according to IFCC (International Federation of Clinical Chemistry and Laboratory Medicine)), no active infection, and written informed consent. Main exclusion criteria were prednisolone therapy > 25 mg/day and planned vaccination of other indication within the study period.

Written informed consent was obtained from all patients prior to study inclusion. The peptide in a water-in-oil emulsion plus an adjuvant was administered subcutaneously four times biweekly. Clinical course, CMVpp65 and IE-1 antigen specific IFNγ release, and CMV-specific CD8^+^ T cells were monitored. Duration of the core study was 56 days, with a follow-up of six months after transplantation followed by an extended observation until months 18 after transplantation. All patients had preemptive CMV therapy after transplantation with careful observations after transplantation including CMV PCR every second week. CMV reactivation was classified as CMV replication, CMV disease (viral detection in body fluid or tissue specimen), and CMV syndrome (two of the following symptoms: fever, malaise, cytopenia, or elevation of hepatic aminotransferases) [20].

The trial was conducted in compliance with Good Clinical Practice (GCP) Guidelines and the Helsinki Declaration of 1975, as revised in 2008.

### 2.2. Patient Samples

Samples were collected from all patients before the first vaccination (T0), prior to each vaccination (T1–T4), and two weeks (T5) after the last vaccination. Peripheral blood mononuclear cells (PBMC) from patients were prepared by Ficoll (Biochrom, Berlin, Germany) separation and tested freshly or after cryopreservation in FCS serum (PAN Biotech, Aidenbach, Germany) containing 10% DMSO (Sigma Aldrich, Steinheim, Germany) and stored in liquid nitrogen.

### 2.3. Vaccine Preparation and Peptide Vaccination

CMVpp65 peptide vaccines were manufactured according to Good Manufacturing Practice (GMP) Guidelines at the GMP Core Facility in Heidelberg as described previously [19]. Three hundred micrograms of CMVpp65-derived peptide (495-NLVPMVATV-503, Bachem Distribution Services, Weil am Rhein, Germany) were emulsified with incomplete Freund’s adjuvant ISA-51, Montanide^®^ (Seppic, Paris, France), and 1400 µL of the emulsion were administered subcutaneously four times in the proximal upper leg. Imiquimod 5% (Aldara^®^, MEDA Pharma, Solna, Sweden) was administered on the skin at the site of the peptide vaccine on the day of vaccination, as well as one day before and two days after peptide vaccination, i.e., four times per peptide administration. All vaccinations had to be performed four times biweekly prior to transplantation. The membrane-bound method was used for sterility testing after validation for bacteria and fungi as required per Ph. Eur. 2.6.1

### 2.4. CMV-Specific Antibodies

CMV-specific antibodies, i.e., the CMV immunoglobulin G index, was assessed by standard assays (Enzygnost anti-CMV IgG/IgM, Siemens, Eschborn, Germany).

### 2.5. CMV Quantitative PCR

DNA was extracted from 200 μL EDTA blood samples and purified using the QIAamp blood kit (QIAGEN, Hilden, Germany) according to the manufacturer’s instructions. A TaqMan real-time PCR assay was performed targeting the UL 86 region in the CMV genome. For quantitative analysis of CMV DNA, 5 μL of extracted nucleic acids were amplified with forward primer CMV1 (5′-CAG CCT ACC CGT ACC TTT CCA-3′) and reverse primer CMV2 (5′-GCG TTT AAT GTC GTC GCT CAA-3′) and detected with the probe 5′-FAM-TTC TAC TCA AAC CCC ACC ATC TGC GC-TAMRA-3′. Additionally, a CMV DNA quantification standard was used threefold in all assays in order to allow quantification of the amplified CMV DNA from patient samples. Quantified CMV DNA was expressed as copies/mL. PCR was performed in a reaction volume of 20 μL with a ready-to-use master mix (Roche Diagnostics, Mannheim, Germany) containing Taq DNA polymerase and dNTPs. Amplification and detection were performed on a LightCycler 480 instrument (Roche Diagnostics, Mannheim, Germany) with a thermocycling profile at 95 °C for 5 min followed by 50 cycles of 95 °C for 5 s and 60 °C for 20 s.

### 2.6. Tetramer Staining for CMV-Specific CD8^+^ T Cells

The frequency of CMV-specific CD8^+^ T lymphocytes was determined by staining with a combination of antibodies (Appendix A) including anti-CD8 FITC antibody (BD, Heidelberg, Germany) and HLA-A2/CMV-tetramer PE as described previously [21]. The acquisition was performed on a LSRII device (BD Biosciences, San Diego, USA) and the analysis was done by BD FACSDiva software (BD bioscience).

A positive immunological response of CMV-specific CD8^+^ T cells was defined as more than 0.1% of CMV-specific CD8^+^ T cells out of the population of the CD8^+^ T cells.

### 2.7. T-Track Assays

Commercially available IFN-γ ELISpot T-Track^®^ CMV (Lophius Biosciences GmbH, Regensburg, Germany) assays were used for the assessment of CMVpp65 and IE-1 antigen specific IFNγ release. T-Track^®^ CMV assays were performed and interpreted according to the manufacturer’s instructions. Briefly, PBMC were stimulated with T-activated^®^ CMV-specific immediate-early 1 (IE-1) and phosphoprotein pp65 (pp65) proteins for 19 h at 37 °C. In the IFN-γ ELISpot assay 2.0 × 10^5^ cells were seeded per well and spot-forming cells (SFC) were enumerated on an automated reader (Bioreader^®^ 5000 Pro-Eα, BIO-SYS GmbH, Karben, Germany). Test results were considered positive if the geometric mean of four replicate spot counts resulting from pp65 or/and IE-1 stimulation was ≥10 SFC/200,000 cells and when the ratio of the geometric means of stimulated to non-stimulated conditions was ≥2.5. SFC counts (geometric mean of four replicates) from unstimulated conditions were subtracted from those of the respective IE-1- and pp65-stimulated conditions, and a minimum SFC value of 0.1 was chosen.

### 2.8. Sample Size Calculation and Statistical Analyses

Patients were enrolled in a two-step 5 + 5 study design to ensure patient safety as appropriate for a clinical phase I study. The first five patients had to complete all four vaccinations as well as the “end of study visit” 14 days after the last vaccination. Solely one patient per day was allowed to receive the first vaccination within the first five patients. If more than one patient had developed toxicity signs above Grade II, the study would have been stopped. If the true rate of toxicity (>Grade II) is 0.50, then the probability that at least two patients out of five suffer from this event and therefore the early termination of the study is about 97%. The probability to find an event of at least Grade II out of ten patients is 99%, when the true rate is 0.5. On the other hand, when the true rate of toxicity is 0.1, the probability to recommend the vaccine is about 93% in the second stage (5–10 patients).

This 5 + 5 design provided the necessary statistical quality for a phase I study.

The results are presented in a descriptive manner with number and percentages for adverse events, and median and ranges for non-parametric data. Parametric data are shown as mean, standard deviation, minimum, and maximum.

## 3. Results

### 3.1. Manufacturing of the Vaccine

All 40 vaccines were individually produced under sterile conditions in full compliance with Good Manufacturing Practice (GMP) requirements. All release criteria including weight and volume, visual control, drop test for consistency, and microscopy for homogeneity of micellular structure were achieved. The content of peptide in emulsion was in the range of 300 µg ± 20% per injection as measured by gas chromatography followed by mass spectrometry using the enantiomer labeling method. In validated post vaccination tests, all vaccines tested sterile according to Ph. Eur. 2.6.1. (Appendix A). The time between vaccine release and acceptance at the hospital for patient application was 9 ± 0.2 min (range 4–15 min). The vaccine was transferred from the GMP unit to the hospital under temperature control. In all 40 vaccine preparations, the temperature was within the prescribed range with a mean increase of 1.93 ± 1.89 °C during transportation.

All patients completed all four vaccinations as well as the 56-day study period before transplantation.

### 3.2. Patients’ Demographics and Clinical Characteristics

Ten patients (six male and four female) were included consecutively at our institution, between February 2015 and May 2016. Table 1 summarizes clinical data of the patients along with dialysis procedures and basic clinical data. All patients were active on the waiting list for renal transplantation. Detailed characteristics of vaccinated patients are shown in Table 2.

### 3.3. Clinical Adverse Events

All ten patients received all four vaccinations. No serious adverse events were detected (Table 3). Altogether, 34 adverse events were documented within the study period, including 13 events without association to vaccination. All 21 vaccination associated side effects (mostly pruritus and pressure pain) were classified as CTC (common toxicity criteria) Grade I reactions of the skin at the site of injection. These side effects resolved without sequels. No other toxicities were observed.

### 3.4. CMV-Specific T Cells and Release

All enrolled patients were CMV IgM/IgG negative prior to vaccination. At baseline, participants showed neither pre-existing CMV-specific CD8^+^ T cells in tetramer-based flow cytometry nor significant (>10/200.000) IFNγ spot-forming cells (SFC).

In five of 10 patients, any immune response was detected by an increase in IFNγ production in the T-Track™ assay and/or an increase of CMV-specific CD8^+^ T cells were observed (Table 4, exemplary Patient 03 in Figure 1). All patients with CMV-specific CD8^+^ T cells presented an increase of CMV-specific effector T cells (Table 4).

None of the patients presented IgG seroconversion after the vaccination.

### 3.5. Clinical Follow-Up

None of the patients developed any CMV disease within the core study period (end of study = Day 56). At the time of analysis, nine patients have had a renal transplantation and one patient has died on the waiting list due to cardiac failure. Renal allograft function was stable in the follow-up period of six months and the following 12-month observation period. All four transplanted patients with immune response did not develop CMV replication in the follow-up period of six months, but one renal allograft recipient presenting with CMVpp65 and IE-1 antigen specific IFNγ release, but missing CMV-specific CD8^+^ T cells, developed CMV disease with a CMV colitis at Month 7. This patient also had a high immunosuppressive load with tacrolimus (Tac) trough levels of about 10 ug/L and mycophenolic acid (MPA) 1000 mg BID. One patient classified as responder received a seronegative organ and did not develop any CMV infection after transplantation including the extended observational period. Four patients classified as non-responders developed CMV replication after transplantation, while the fifth non-responding patient received a CMV negative organ. Without any inoculation with CMV, of course no reactivation could occur. Two of these four patients presented with CMV syndrome, one with CMV disease (pneumonia) and one with CMV replication only. The last patient had presented with very small CMVpp65 and IE-1 antigen specific IFNγ release after vaccination, but missing CMV-specific CD8^+^ T cells. Due to very small immune response, the patient was primarily classified as non-responder. Further information is given in Table 5.

## 4. Discussion

In this investigator-initiated phase I study, safety and feasibility of a specific CMV peptide vaccination in end-stage renal disease patients on preparation to renal transplantation was assessed for the first time.

In the most recent phase I study, the present CMVpp65-derived peptide was used to vaccinate ten patients receiving an allogeneic stem cell graft from CMV-seronegative donors with encouraging results [19]. As in the current study, vaccination was well tolerated. Seven of nine patients cleared CMVpp65 antigenemia after four vaccinations. In that study, only one patient received prophylactic vaccination, but this patient did not develop antigenemia. This observation underlines our prophylactic approach in end-stage renal patients on the transplant waiting list.

In the present study, the novel method of peptide vaccination was used. Conventional vaccine strategies have been highly efficacious for several decades in reducing mortality and morbidity due to infectious diseases. However, conventional vaccines, such as those that include whole organisms or large proteins, appear to have some adverse side effects due to inclusion of unnecessary antigenic load [22]. A high antigenic load might complicate the vaccination due to induction of allergenic responses. Peptide vaccination is an attractive alternative strategy that relies on usage of short peptide fragments to engineer the induction of highly targeted immune responses. On the other side, peptide vaccines are often weakly immunogenic and require adjuvants. In the present study, a specific CMV peptide vaccine was used in combination with incomplete Freund’s Adjuvant (Montanide^®^) and local application of imiquimod (Aldara^®^ 5% cream). Both adjuvants had been used successfully and safely in earlier studies.

In our present study on further kidney transplant recipients, patients received four vaccinations as per protocol. No serious adverse drug reactions or serious adverse events were detected. All 19 side effects (mostly pruritus and pressure pain) were classified as CTC (common toxicity criteria) Grade I reactions of the skin at the site of injection. These side effects resolved without sequels. No other toxicities were observed (Table 3).

All enrolled patients were CMV IgM/IgG negative prior to vaccination. Five of the 10 patients (50%) mounted any immune response. Four patients developed CMV-specific effector T cells and one patient developed significant IE-1- and pp65-specific spot formatting cells in the IFN-γ ELISpot assay only, all five patients were classified as CMV peptide vaccination responders (Table 4 and Figure 1).

These results are corresponding to other studies on vaccination response in end-stage renal disease patients. In dialysis patients, response rates between 35% and 67% are reported after hepatitis B and influenza vaccination depending on the type of vaccine and number of applied dosages as well as additional boost vaccinations [23,24,25,26].

Protective immunity can be induced by the formation of protective antibodies which requires an effective cross-linking of B cell receptors on B cells stimulating B cell affinity maturation. Monomeric peptide vaccines are rather poorly immunogenic with regard to B cell stimulation and antibody formation [27]. This is consistent with our observation in the study.

Epitope-specific T cell stimulation is another mechanism by which vaccines can induce protective immunity. Peptides can be presented by antigen-presenting cells (APCs) on the peptide binding groove of Class I or II major histocompatibility complexes (MHCs) to the T cell receptor (TCR) of T cells and can lead to a peptide-specific T cell clone expansion. For T-cell epitopes, immunodominance is an important consideration for peptide vaccine design. Moreover, the vaccine is injected with an adjuvant (e.g., Freund’s adjuvant) to boost the immune response by increasing the half-life of the epitope by decreasing the susceptibility to proteolytic degradation.

Next to the optimal vaccine, the immune system of the patient is of major relevance for the immune response. Patients with end-stage renal disease have an altered immune system with an impaired innate and adaptive immune response. Monocytes and monocyte-derived dendritic cells as the key players for antigen presentation in the vaccination strategy have been shown to display decreased endocytosis and impaired maturation in end-stage renal disease [28]. This might be the major reason for an impaired immune response to an active vaccination strategy in end-stage renal disease patients. It is known that cellular immunity through effector CD4^+^ and CD8^+^ T cells play a critical role for controlling CMV replication after transplantation [29]. Therefore, the novel T-Track^®^ CMV IFN-γ ELISpot assay was used to measure sensitively the response of a large spectrum of clinically relevant CMV-reactive effector cells including T helper cells, cytotoxic T lymphocytes, as well as natural killer and natural killer T-like cells via bystander activation to immediate early-1 (IE-1) and phosphoprotein 65 (pp65) antigens [30]. IE-1 specific reaction is supposed to show immediate immune response, whereas the development of pp65 specific activation is expected to represent long-term immune response. In three patients, a significant increase of IE-1 and pp65-specific SFC could be detected after vaccination.

Nickel et al. described an association between CMV disease with low IE-1-specific T-cell frequencies in a pilot study on renal allograft recipients [31]. In a previous observational study, it has been shown that IE-1 CMV specific T cell frequencies before transplantation could help to discriminate those patients with no need of CMV prophylactic treatment form those in whom prophylaxis is indicated [32]. Intrinsic impairment of IE-1 specific T cell response but not pp65-specific T cells was associated with post-transplant CMV infection as well as CMV disease. In addition, in this previous study only patients with adequate pre-transplant anti-IE-1-specific T cell frequencies were at significant low-risk for CMV infection demonstrating that although CMV triggers both humoral and cellular immune responses, especially the cellular response directed to IE-1 CMV antigen seemed to be important for post-transplant viral replication control. In parallel to our study, Bestard et al. [32] also showed that patients who had not experienced CMV infection showed a significantly lower T cell response when compared to patients with an earlier infection period. Concerning IE-1-specific T cell response the threshold of 7 spots in 3 × 10^5^ showed a high negative predictive value of 95.7%.

None of the patients presented IgG seroconversion after the vaccination. Humoral immunity after primary viral infection is long-lasting. However, the contribution of antibodies (as assessed by standard serology) towards protection against CMV replication in transplant recipients is questionable [33].

After transplantation, immunosuppression may interfere with CMV immune response [29]. T-cell depleting agents increased the risk for CMV infection due to direct depletion of functional CMV-specific T cells or induction of proinflammatory cytokine release which is involved in the activation of latent CMV. Mycophenolic acid blocking activated lymphocytes may facilitate CMV infection especially in high doses. While T-cell depleting agents were prohibited in this study, the use of mycophenolic acid might have influenced even the after vaccination existing cellular immune response. However, none of the vaccinated patients with immune response suffered from clinically overt CMV disease within the first six months, whereas one patient who received a renal allograft more than four years after vaccination developed CMV disease seven months after transplantation. This patient presented only with CMVpp65 and IE-1 antigen specific IFNγ release, but missing CMV-specific CD8^+^ T cells (Table 5). In addition, this patient had a high immunosuppressive load in the weeks prior to CMV disease. Forty percent of the patients showed CMV-specific CD8^+^ T cell responses elicited by these prophylactic vaccinations. All responders did never experience CMV reactivation in the 18 months after transplantation, while all non-responders reactivated. In summary, prophylactic CMV specific peptide vaccination before kidney transplantation induces a T cell mediated response and therefore may prevent CMV infection after transplantation. Since immune response among the patients’ is heterogenous and sometimes unpredictable, a pre-vaccination immune test as the CD4 T cell count might be helpful to detect patients with potential response to vaccination. Because of the small numbers of patients, the correlation between CMV-specific T-cell reactivity and vaccine response requires further investigation. Further studies with an increased patient number and multi-center assessment are necessary to confirm our results. Future vaccines might also include more viral antigen peptides including HLA class II antigens.

## Figures and Tables

**Figure 1 vaccines-09-00133-f001:**
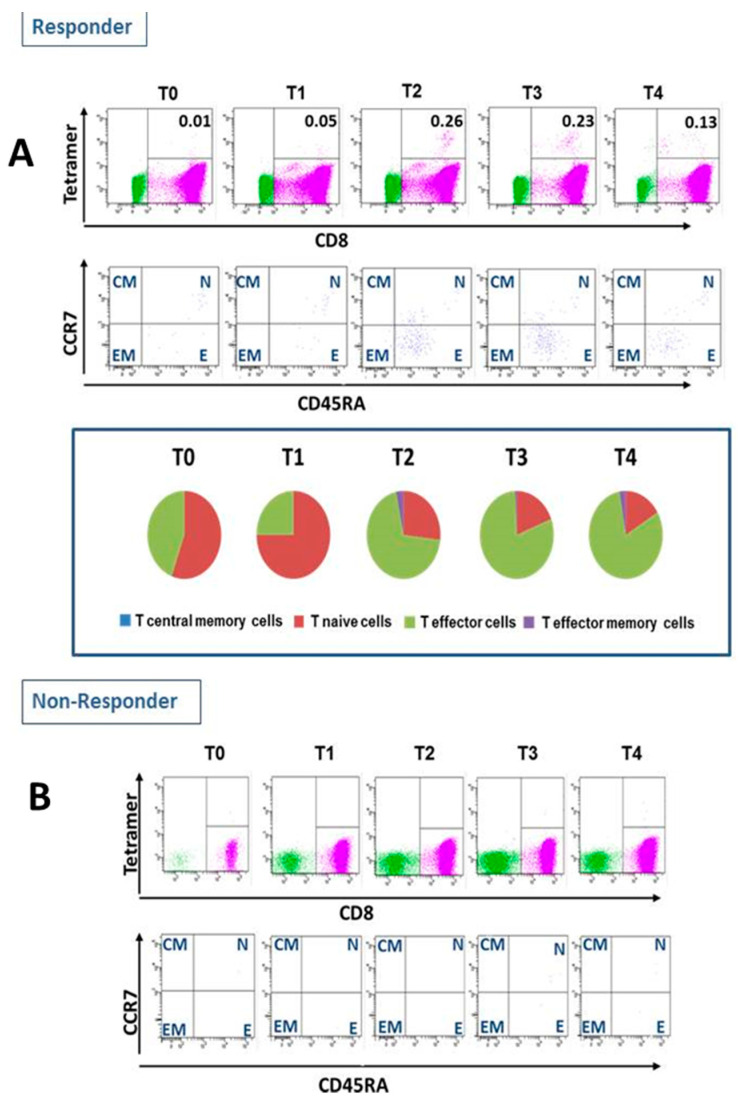
Cytomegalovirus(CMV)-specific CD8^+^ T cells frequency and subsets in patients receiving CMV peptide vaccination. (**A**): Example of a patient(#003) with positive immune response to cytomegalovirus (CMV)-specific peptide vaccination (responder). a. Classical AMV-specific ab CD8^+^ T cells were assessed with tetramer-based flow cytometry. b. Change in T-cell subsets. (**B**): Example of patient (#001) with negative immune response to cytomegalovirus(CMV)-specific peptide vaccination (non- responder). a. Classical AMV-specific ab CD8^+^ T cells were assessed with tetramer-based flow cytometry.

**Table 1 vaccines-09-00133-t001:** Demographic data of study participants.

Parameter		Result
Gender (female/male)	n (%)	4/6 (40/60)
Age	(years), mean ± SD	49.7 ± 12.7
Body Mass Index	(kg/m^2^), mean ± SD	24.1 ± 2.1
Renal disease		
Glomerulonephritis	n (%)	5 (50)
ADPKD	n (%)	1 (10)
Alport Syndrome	n (%)	1 (10)
Nephrocalcinosis	n (%)	1 (10)
Analgesic Nephropathy	n (%)	1 (10)
Unknown (shrunken kidney)	n (%)	1 (10)
Pre-emptive transplantation	n (%)	1 (10)
Dialysis		
Hemodialysis	n (%)	7 (70)
Peritoneal Dialysis	n (%)	2 (20)
Time to Renal Replacement Therapy	(months), mean ± SD	84.6 ± 29.4

**Table 2 vaccines-09-00133-t002:** Clinical characteristics of vaccinated patients.

Pat #	Gender	Age (Years)	BMI (kg/m^2^)	Underlying Renal Disease	Type of RRT	Time on RRT (Months)	Transplant Program	Karnofsky Index	Blood Pressure (mmHg)	Creatinine (mg/dL)	Albumin (g/L)
01	F	46	26.9	ADPKD	Pre-emptive	0	Living	0.9	120/70	5.90	44.0
02 *	M	67	26.7	GN	HD	39	ESP	0.8	162/93	6.95	47.5
03	M	25	21.1	Alport syndrome	HD	115	ETKAS	0.9	150/95	9.18	40.9
04	F	45	26.7	unknown	HD	90	ETKAS	0.9	124/86	9.40	42.9
05	M	59	22.8	GN	HD	125	ETKAS	0.9	130/80	7.31	44.3
06	M	39	22.5	GN	PD	75	ETKAS	0.9	140/90	12.5	38.2
07	M	57	22.2	GN	HD	42	ETKAS	1	115/84	6.76	44.3
08	F	45	25.1	Analgesic nephropathy	PD	80	ETKAS	0.7	130/85	11.4	37.1
09	F	65	23.0	Nephro-calcinosis	HD	98	ETKAS	1	180/80	8.17	38.9
10	M	49	24.1	GN	HD	97	ETKAS	1	110/75	12.1	42.7

ADPKD, autosomal-dominant polycystic kidney disease; BMI, body mass index; ESP, Eurotransplant Senior Program; ETKAS, Euro Transplant Kidney Allocation System; F, female; GN, glomerulonephritis; HD, hemodialysis; M, male; PD, peritoneal dialysis; RRT, renal replacement therapy; * Patient 02 had already undergone a preceding kidney transplantation.

**Table vaccines-09-00133-t003a:** (a)

Pat #	Number of Events	Inflammation	Swelling	Pruritus	Pain	Hematoma	Fatigue
01	4	1	1	1			1
02	1		1				
03	2	1		1			
04	1			1			
05	2			2			
06	2		1		1		
07	3			1	1	1	
08	4	1		1	1	1	
09	1					1	
10	1						1

**Table vaccines-09-00133-t003b:** (b)

Pat #	Number of Events	Respiratory Tract Infection	Gastro-Intestinal Infection	Muscle Cramps	Hyperkalemia	Hypotension	Pollinosis	Renal Cyst Bleeding	Abrasions after Bicycle Accident
01	2						1	1	
02									
03	2	1							1
04	4	3			1				
05	1	1							
06	1			1					
07									
08	1		1						
09									
10	2				1	1			

**Table 4 vaccines-09-00133-t004:** Virus status and immunological responses up to the end of the study. Prior to vaccination, none of the 10 patients had experienced a CMV infection. All patients had a negative serostatus for CMV. No CMV-neutralizing antibodies were found in the serum. IFT for CMVpp65 and qPCR for CMV tested native in all patients.

Pat #	T-Track^®^ CMV (Antigen-Specific SFC/200.000 Lymphocytes) *		CMV-Specific CD8+ T Cells % ^#^ before → after Vaccination	CMV-Specific Effector T Cells (EM) % before → after Vaccination	Any CMV Peptide Immune Reaction (FACS and/or Elispot)
	IE-1^+^ before → after Vaccination	pp65^+^ before → after vaccination			
01	1 → 1	0 → 1	0 → 0	NA	No
02	2 → 14	1 → 13	0 → 0	NA	No
03	1 → 42	0 → 12	0 → 0.3	36 → 87	Yes
04	3 → 1	2 → 3	0 → 0	NA	No
05	4 → 24	0 → 4	0 → 0.1	33 → 86	Yes
06	1 → 20	1 → 10	0 → 0.1	31 → 72	Yes
07	0 → 104	0 → 50	0 → 0	NA	Yes
08	4 → 2	2 → 4	0 → 0	NA	No
09	1 → 1	1 → 0	0 → 0.1	0 → 91	Yes
10	1 → 1	1 → 2	0 → 0	NA	No

CMV, cytomegalovirus; PCR, polymerase chain reaction; IFNy, interferon y, NA, not applicable. * T-Track^®^ CMV: positive >10 antigen-specific spot-forming colonies (SFC) per 200,000 lymphocytes. ^#^ Tetramer staining: (0) Negative → 0.0%; (1) low → 0.1–1.0% tetramer+ CD8^+^ T cell; (2) medium → 1–5%; (3) high → >5%.

**Table 5 vaccines-09-00133-t005:** Virus status, immunological responses and clinical outcome within the follow-up after transplantation.

Pat #	Patient Outcome	Time from Vaccination to Transplantation (Months)	CMV Status	Prophylactic CMV Treatment	Immunosuppressive Regimen	S-Creatinine (mg/dL), Month 6	CMV Replication/Disease until Month 18 after tx	CMV Specific T Cell Response
01	Living kidney tx *	7	D+/R-	pre-emptive	plasmapheresis, rituximab, ATG, steroids, MPA, Tac ^+^	1.20	Yes (CMV syndrome) (month 18 after tx)	No
02	Deceased kidney tx	2	D+/R-	pre-emptive	basiliximab, steroids, MPA, Tac	1.54	Yes (CMV replication) (month 5 after tx)	No
03	Deceased kidney tx	43	D+/R-	pre-emptive	basiliximab, steroids, MPA, CsA	2.05	No	Yes
04	Deceased kidney tx *	46	D+/R-	pre-emptive	ATG, steroids, MPA, Tac	1.59	Yes (CMV disease) (month 5 after Tx)	No
05	Deceased kidney tx	17	D+/R-	pre-emptive	ATG, steroids, MPA, Tac	1.94	No	Yes
06	Deceased kidney tx	48	D-/R-	pre-emptive	basiliximab, steroids, MPA, Tac	1.59	No	Yes
07	Deceased kidney tx	53	D+/R-	pre-emptive	basiliximab, steroids, MPA, Tac	1.28	Yes (CMV disease) (month 7 after tx)	Yes
08	Deceased kidney tx	4	D+/R-	pre-emptive	basiliximab, steroids, MPA, CsA	1.67	Yes (CMV syndrome) (month 12 after tx)	No
09	Died from cardiac failure on waiting list	NA	NA	NA	NA	NA	NA	Yes
10	Deceased kidney tx	18	D-/R-	pre-emptive	basiliximab, steroids, MPA, CsA	1.75	No	No

CMV, cytomegalovirus; CsA, Ciclosporin A; IFNy, interferon y; MPA, mycophenolate acid; NA, not applicable; PCR, polymerase chain reaction; Tac, tacrolimus; Tx, transplantation.

## Data Availability

The data presented in this study are available on request from the corresponding author. Data are not publicly available due to the national law.

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
