# Peer review of "Peptide Vaccination against Cytomegalovirus Induces Specific T Cell Response in Responses in CMV Seronegative End-Stage Renal Disease Patients"

_vaccines, 2021, doi:10.3390/vaccines9020133_

Round 1

Reviewer 1 Report

In this manuscript, Sommerer et al examine the use of a CMVpp65 peptide vaccination strategy in a phase I clinical trial of ten patients with end stage renal disease. Specifically, they focus on the safety of the vaccine and how well it’s tolerated, whilst also looking at its efficacy in generating CMV-specific cellular immunity and preventing CMV reactivation. This approach is similar to work previously published by the group in a different clinical context and the effect of the vaccine was encouraging. In all patients that responded to vaccine (50%), none of these patients displayed any CMV reactivation post-transplantation. The data is intriguing and is robust enough to a certain degree although I do have some concerns about the presentation of the data and the conclusions made that need improvement or to be addressed. My comments are as follows:-

Comments:

  1. In patient #2, pp65-specific CD8+ T cells were picked up by ELISpot to similar levels seen in other patients (e.g. #3), but this patient was deemed to have generated no obvious “CMV peptide immune reaction”. More importantly, this patient displayed CMV-reactivation post-transplantation. Why wasn’t this considered when reporting the results? Additionally, for patient #7, the data in Table 4 and 5 does not match with respect to evidence of CMV-specific T cell responses. Table 4 says yes, Table 5, says no. This is important as this patient also reactivated post-Tx. Is there a reason for this discrepancy? Thus, arguably, 2/10 patients showed CMV reactivation despite generating CMV-specific cellular immunity. Therefore, this affects all the conclusions made about efficacy of the vaccine.

  1. The production of CMV-specific CD8+ T cells post-vaccination only worked in 5/10 patients (author’s words), despite all of them being HLA-A*02+. The reasons why it did not work in the other 5 patients are not discussed properly. The authors need to address this.

  1. The data in Figure 1 is, visually, of poor quality which needs improvement to be more legible to the reader. That aside, the phenotypic comparison in Fig 1A appears to be on tetramer+ events but this is not defined clearly. If so, it seems that the quadrant gating is not correct as naïve T cells have much higher expression of CD45RA than effector cells (which are better referred to as TEMRA cells). Setting the gates properly would show that more of the cells are TEM cells. This needs to be corrected and could be helped by using all CD8s as a background for setting the gates.

  1. The data presented in Table 4 in many ways is hard to understand. There are many columns there are two numbers compared with an arrow between them. Do these numbers compare different time points? If so, this should be indicated. Also, similarly the percentages of CMV-specific effector cells are confusing. If the first number is pre-vaccination, then surely this number should be zero in all since all patients were seronegative. Thus, could this data reflect the global CD8 population? Again, what are these time points? This whole table needs to be cleaned up to be suitable for publication.

  1. The examination of CMV-specific CD8+ T cells by flow cytometry appears to use a strategy of just using anti-CD8 and tetramer bound to PE but there is no obvious use of a CD3 antibody to define all T cells prior to this. Given that CD8 can be found on other cell types, including dendritic cells, it would be good to know that CD3 was used here.

  1. Part of the work was to look at CMV-specific humoral immunity but this data was not shown (page 6). Surely it should be, even though the results proved there was no seroconversion post vaccination?

Minor comments:

  1. Table 1: The numbers in brackets are not defined. I believe these are percentages but this should be indicated
  2. There are a few grammar issues in the paper. Please check through and correct.

Author Response

Reviewer 1
Comments and Suggestions for Authors
In this manuscript, Sommerer et al examine the use of a CMVpp65 peptide vaccination strategy in a phase I clinical trial of ten patients with end stage renal disease. Specifically, they focus on the safety of the vaccine and how well it’s tolerated, whilst also looking at its efficacy in generating CMVspecific cellular immunity and preventing CMV reactivation. This approach is similar to work
previously published by the group in a different clinical context and the effect of the vaccine was encouraging. In all patients that responded to vaccine (50%), none of these patients displayed any CMV reactivation post-transplantation. The data is intriguing and is robust enough to a certain degree although I do have some concerns about the presentation of the data and the conclusions made that need improvement or to be addressed. My comments are as follows:-

Thank you very much for your assessment. We greatly appreciate your time and effort for our manuscript.

Comments:
1. In patient #2, pp65-specific CD8+ T cells were picked up by ELISpot to similar levels seen in other patients (e.g. #3), but this patient was deemed to have generated no obvious “CMV peptide immune reaction”. More importantly, this patient displayed CMV-reactivation post-transplantation. Why wasn’t this considered when reporting the results? Additionally, for patient #7, the data in Table 4 and 5 does not match with respect to evidence of CMV-specific T cell responses. Table 4 says yes, Table 5, says no. This is important as this patient also reactivated post-Tx. Is there a reason for this discrepancy? Thus, arguably, 2/10 patients showed CMV reactivation despite generating CMVspecific cellular immunity. Therefore, this affects all the conclusions made about efficacy of the
vaccine.

Answer:
Thank you very much for your constructive comments.
We classified the immunological responses of patients according to the results of the Elispot assay and the flow cytometry data. Positive results in both assays, Elispot and FACS, were classified as positive immune reactions. Clearly positive results in only one of these assays were classified as positive, low positive results in only one of these assays were rather considered as unspecific reactions.

According to this specification patients #2, #5, #6, #7 and #9 were classified as positive and patients #1, #2, #4, #8 and #10 were classified as negative for CMV-specific peptide immune responses. We corrected the tables 4 and 5 accordingly.
Patient #7, who has been classified as positive for CMV-specific immune response by presenting CMVpp65 and IE-1 antigen specific IFNgamma release, has experienced CMV disease with CMV colitis at month 7 after kidney transplantation and more than 4 years after vaccination. This might be due to the highly immunosuppressive regimen with tacrolimus (Tac) trough levels of about 10 ug/L and mycophenolic acid (MPA) 1000 mg BID in the weeks prior to the CMV reactivation.

In summary, only 1/10 patients showed a CMV reactivation despite a CMV specific immune response at month 7 after kidney transplantation due to a highly immunosuppressive regimen.

2. The production of CMV-specific CD8+ T cells post-vaccination only worked in 5/10 patients (author’s words), despite all of them being HLA-A*02+. The reasons why it did not work in the other 5 patients are not discussed properly. The authors need to address this.

Answer:
That is an important aspect. We have incorporated the following text into the discussion:
“Protective immunity can be induced by the formation of protective antibodies which requires an effective cross-linking of B cell receptors on B cells stimulating B cell affinity maturation. Monomeric peptide vaccines are rather poorly immunogenic with regard to B cell stimulation and antibody formation (Malonis RJ et la, Peptide-Based Vaccines: Current Progress and Future Challenges, Chem. Rev. 2020, 120, 3210−3229). This is consistent with our observation in the study.
Epitope-specific T cell stimulation is another mechanism by which vaccines can induce protective immunity. Peptides can be presented by antigen-presenting cells (APCs) on the peptide binding groove of class I or class II major histocompatibility complexes (MHCs) to the T cell receptor (TCR) of T cells and can lead to a peptide-specific T cell clone expansion. For T-cell epitopes, immunodominance is an important consideration for peptide vaccine design. Moreover, the vaccine is injected with an adjuvant (f.e. Freund`s adjuvant) to boost the immune response by increasing the half-life of the epitope by decreasing the susceptibility to proteolytic degradation.
Next to the optimal vaccine the immune system of the patient is of major relevance for the immune response. Patients with end-stage renal disease have an altered immune system with an impaired innate and adaptive immune response. Monocytes and monocyte-derived dendritic cells as the key players for antigen presentation in the vaccination strategy have been shown to display decreased endocytosis and impaired maturation in end-stage renal disease (Kator S, Aspects of immune dysfunction in end-stage renal disease, Clin J Am Soc Nephrol. 2008). This might be the major reason for an impaired immune response to an active vaccination strategy in end-stage renal disease patients.”

3. The data in Figure 1 is, visually, of poor quality which needs improvement to be more legible to the reader. That aside, the phenotypic comparison in Fig 1A appears to be on tetramer+ events but this is not defined clearly. If so, it seems that the quadrant gating is not correct as naïve T cells have much higher expression of CD45RA than effector cells (which are better referred to as TEMRA cells). Setting the gates properly would show that more of the cells are TEM cells. This needs to be corrected and could be helped by using all CD8s as a background for setting the gates.

Answer:
Thank you very much for this important point and your suggestion to reevaluate our data. We revised Figure 1 and and accordingly Table 4 as you suggested.

4. The data presented in Table 4 in many ways is hard to understand. There are many columns there are two numbers compared with an arrow between them. Do these numbers compare different time points? If so, this should be indicated. Also, similarly the percentages of CMV-specific effector cells are confusing. If the first number is pre-vaccination, then surely this number should be zero in all since all patients were seronegative. Thus, could this data reflect the global CD8 population? Again, what are these time points? This whole table needs to be cleaned up to be suitable for publication.

4. The data presented in Table 4 in many ways is hard to understand. There are many columns there are two numbers compared with an arrow between them. Do these numbers compare different time points? If so, this should be indicated. Also, similarly the percentages of CMV-specific effector cells are confusing. If the first number is pre-vaccination, then surely this number should be zero in all since all patients were seronegative. Thus, could this data reflect the global CD8 population? Again, what are these time points? This whole table needs to be cleaned up to be suitable for publication.

Answer:
We have added additional information “before – after vaccination” into the table for better understanding. The data reflect just the CMV-specific CD8+ T cells and the subpopulation T(em) specific T cells.

5. The examination of CMV-specific CD8+ T cells by flow cytometry appears to use a strategy of just using anti-CD8 and tetramer bound to PE but there is no obvious use of a CD3 antibody to define all T cells prior to this. Given that CD8 can be found on other cell types, including dendritic cells, it would be good to know that CD3 was used here.

Answer:
Our gating strategy comprises the gating of viable CD3+, CD8+ and CMV Tetramer positive lymphocyte populations followed by the analysis of CCR7 and CD45RA stained subpopulations. Monocytes, B cells and dead cells were excluded in the dump channel (CD14, CD19, 7AAD).
Gating strategy: see attachment

6. Part of the work was to look at CMV-specific humoral immunity but this data was not shown (page 6). Surely it should be, even though the results proved there was no seroconversion post vaccination?

Answer:
All enrolled patients were CMV IgM/IgG negative prior to vaccination and none of the patients showed a seroconversion. Therefore we think that a presentation of the humoral immunity data might not be of interest.
However, we included an explanation for this into the discussion part:
“Protective immunity can be induced by the formation of protective antibodies which requires an effective cross-linking of B cell receptors on B cells stimulating B cell affinity maturation. Monomeric peptide vaccines are rather poorly immunogenic with regard to B cell stimulation and antibody formation (Malonis RJ et la, Peptide-Based Vaccines: Current Progress and Future Challenges, Chem. Rev. 2020, 120, 3210−3229). This is consistent with our observation in the study. “

Answer:
All enrolled patients were CMV IgM/IgG negative prior to vaccination and none of the patients showed a seroconversion. Therefore we think that a presentation of the humoral immunity data might not be of interest.
However, we included an explanation for this into the discussion part:
“Protective immunity can be induced by the formation of protective antibodies which requires an effective cross-linking of B cell receptors on B cells stimulating B cell affinity maturation. Monomeric peptide vaccines are rather poorly immunogenic with regard to B cell stimulation and antibody formation (Malonis RJ et la, Peptide-Based Vaccines: Current Progress and Future Challenges, Chem. Rev. 2020, 120, 3210−3229). This is consistent with our observation in the study. “

Minor comments:
1. Table 1: The numbers in brackets are not defined. I believe these are percentages but this should be indicated
There are a few grammar issues in the paper. Please check through and correct.

Answer:
The numbers in brackets have been defined. We reorganized Table 1 and corrected the grammar issues.

Reviewer 2 Report

Manuscript submitted by Sommerer and colleagues describes a small clinical phase I trial by including 9 kidney transplant recipients. Authors tested efficacy of CMVpp65 peptide vaccination among mostly CMV positive donor and negative recipient (D+ /R-) end-stage renal disease patients. Authors conclude that peptide vaccine was well tolerated and patients with vaccine-induced acquired immunity were protected from CMV reactivation. 

The concept of CMV protection by peptide based vaccination is promising. However, manuscript lacks several key details, making it difficult to draw conclusions. Authors are encouraged to revise the manuscript based on the comments below and resubmit it.

  1. Abbreviation "D+R-" should be elaborated in line 52.
  2. Statement in line 63-64 is not clear. 
  3. What parameters were used to define "end-stage renal disease patients" in line 81?
  4. Statement "liver function tests below the three fold of the normal upper values " could be replaced by providing real values and details of which test was used to measure liver function tests. These details should be included in the study design.
  5. Statement in line 87-88, need rationale and details of how water-in-oil emulsion plus adjuvant (provide details) combination was finalized? What concentration of peptide was used and how four doses of the injection were calculated? 
  6. Also, details on use of immunosuppresant are missing. 
  7. Please provide the gene accession number to which sequence of CMVpp65-derived peptide corresponds.
  8. What volume of injected mixture of peptide+Freund's adjuvant was used? (Line 109)
  9. Please provide details of antibody (line 118). Clone number, catalog number, and manufacturer. 
  10. For ELISpot assay, how many cells were seeded per well?
  11. In Line 126 and 127, what does the asterisk mark represent?
  12. The statement, "following experienced statistician advice ", in line 148, is not scientific. Please provide what tools were used to determine size of the study, e.g. power analysis etc?
  13. Section 2.7, please provide details of what statistical tools were used? e.g. ANOVA etc. 
  14. Section 3.1, a representative data for the release criteria should be shown (it could be provided as supplementary data).
  15. Line 162. Brief description of method used to test vaccine sterility should be provided in Materials and Methods section.
  16. Line 164. Under what temperature vaccine was transferred? What was the prescribed range?
  17. Line 167, it is not clear if patients were under constant observation (hopitalized) or stay home?
  18. Data in table 3 could be provided per patient, so that correlation with existing disease status could be established.
  19. Section 3.3, second paragraph is not clear. If immune response was detected or not?
  20. Table 4. Data for SFC should be presented as SFC/ million lymphocytes.
  21. Please clearly explain the data shown. In the 2nd column, Is this SFC at T0 (time point before vaccination) vs T5 (time point after transplantation or vaccination)?
  22. Details of qPCR for CMV should be provided. 
  23. In column 5, "Yes" summarizes results from FACS or ELISPOT or both?
  24. For Fig 1, please provide a gating strategy.  Also, explain why CD8 T cell frequency is so high, and CD8 negative is so low? Generally CD8- population is higher than CD8+. 
  25. Indicate, on the basis of flow results, which patient was responder?
  26. Since only 5 patients were responders, it is not clear how CMV was controlled in other patients? 
  27. How can the rate of responders be enhanced? Future strategy should be discussed

Author Response

Reviewer 2

Comments and Suggestions for Authors

Manuscript submitted by Sommerer and colleagues describes a small clinical phase I trial by including 9 kidney transplant recipients. Authors tested efficacy of CMVpp65 peptide vaccination among mostly CMV positive donor and negative recipient (D+ /R-) end-stage renal disease patients. Authors conclude that peptide vaccine was well tolerated and patients with vaccine-induced acquired immunity were protected from CMV reactivation. 

The concept of CMV protection by peptide based vaccination is promising. However, manuscript lacks several key details, making it difficult to draw conclusions. Authors are encouraged to revise the manuscript based on the comments below and resubmit it.

Thank you very much for your assessment. We greatly appreciate your time and effort for our manuscript.

  1. Abbreviation "D+R-" should be elaborated in line 52.

Answer:

We spelled this abbreviation out on page 2, paragraph 3.

„Antiviral prophylaxis is standard of care at least in patients with CMV high-risk constellation i.e. donor CMV-seropositive/ recipient seronegative (D+R-).“

  1. Statement in line 63-64 is not clear.

Answer:

This sentence has been corrected for better understanding as follows: „Recently, a first phase I trial in patients after hematopoietic stem cell transplantation has shown that this CMV peptide vaccination was safe, well tolerated and efficacious [19].“

  1. What parameters were used to define "end-stage renal disease patients" in line 81?

Answer:

Usually patients on renal replacement therapy as hemodialysis or peritoneal dialysis or patients very near to start renal replacement therapy are called “endstage renal disease (ESRD)” patients. It is a ordinary terminus in nephrology. There is a typo in ESRD which is corrected in the revised version.

  1. Statement "liver function tests below the three fold of the normal upper values" could be replaced by providing real values and details of which test was used to measure liver function tests. These details should be included in the study design.

Answer:

This information is given in the Method section of the revised manuscript: “ liver function tests (alanine aminotransferase, aspartate aminotransferase, alkaline phosphatase, gamma-glutamyl-transpeptidase) below the threefold of the normal upper values (ultraviolet test according to IFCC (International Federation of Clinical Chemistry and Laboratory Medicine))“.

  1. Statement in line 87-88, need rationale and details of how water-in-oil emulsion plus adjuvant (provide details) combination was finalized? What concentration of peptide was used and how four doses of the injection were calculated?

Answer:

Manufacturing of the vaccine was elaborated on 2.3: 300 μg of CMVpp65-derived peptide (495-NLVPMVATV-503, Bachem Distribution Services, Weil am Rhein, Germany) was emulsified with incomplete Freund’s adjuvant ISA-51, Montanide® (Seppic, Paris, France) as described earlier in ref. 19. “Briefly, the CMVpp65495-503NLVPMVATV peptide (N9V; HLA-A2) lysate was resolubilized in DMSO and further dissolved in phosphate buffered saline with ethylene-diamine-tetraacetic acid (PBS /EDTA). Thereafter the mixture was drawn into a syringe with a total volume of 2,800 μl. An equal volume of ISA 51/Montanide™, was drawn into a second syringe, then both syringes were connected and a water-in-oil emulsion was produced by mixing the components slowly followed by fast mixing. The quality of the vaccine was determined by an optical control under the microscope and a control of the correct viscosity by dripping one drop of the mixture on sterile PBS. Thereafter the emulsified vaccine was filled into a fresh syringe for the patient and controls were taken for sterility, determination of peptide content and retention samples. Aliquots of the vaccine were sent to external laboratories for the assessment of sterility (L+S AG, Bad Bocklet, Germany) and the content of

peptide in emulsion (C.A.T. GmbH, Tübingen, Germany).

All release criteria like weight and volume, visual control, drop test for consistency, microscopy

for homogeneity of micellular structure were fulfilled. In validated post vaccination tests all vaccines tested sterile according to Ph. Eur. 2.6.1, and the content of peptide in emulsion was in the range of 300 μg +/- 20% per injection as measured by gas chromatography followed by mass spectrometry

using the enantiomer labeling method.”

All vaccines were administered subcutaneously four times at a volume of 1,400 µl in the proximal upper leg in a two weeks’ interval. At each time point, vaccines were freshly manufactured.

  1. Also, details on use of immunosuppresant are missing.

Answer:

The immunosuppressive regimen of each patient has already been provided in Table 5.

  1. Please provide the gene accession number to which sequence of CMVpp65-derived peptide corresponds.

Answer:

The gene accession number of CMVpp65-derived peptide (495-NLVPMVATV-503) is 5D2N_I.

  1. What volume of injected mixture of peptide+Freund's adjuvant was used? (Line 109)

Answer:

We have added the volume of injected mixture of peptide plus Freund's adjuvant of 1,400 µl in the Material and Methods part under 2.3. on page 3.

  1. Please provide details of antibody (line 118). Clone number, catalog number, and manufacturer.

Answer:

Samples were stained with the following antibodies:

CD14 PerCP (clone: HCD14, Cat. No.: 325632, Biolegend)

CD19 PerCP(clone: HIB19, Cat. No.: 302228, Biolegend)

7AAD (Cat. No.: 559925, BD)

CD3 V450 (clone: UCHT1, Cat. No.: 560366, BD)

CCR7 PE-Cy7 (clone: 3D12, Cat. No.: 25-1979-42, eBioscience)

CD45RA APC (clone: HI100, Cat. No.: 304112, Biolegend)

CD8 FITC (clone: SK1, Cat. No.: 344704, Biolegend)

Tetramer PE (monomer from NIH, PE-streptavidin: Cat. No.: 405204, Biolegend)

CD14 PerCP, CD19 PerCP and 7AAD belong to the dump channel.

We have added a Supplementary Table 1 with detailed information of the antibodies.

  1. For ELISpot assay, how many cells were seeded per well?

Answer:

In the IFN-γ ELISpot assays 2.0 x 105 cells were seeded per well. This is written in the Method part 2.7., page 4, of the revised version.

  1. In Line 126 and 127, what does the asterisk mark represent?

Answer:

We deleted the respective asterisks.

  1. The statement, "following experienced statistician advice ", in line 148, is not scientific. Please provide what tools were used to determine size of the study, e.g. power analysis etc?

Answer:

Extensive description of the type of phase I study is provided (section 2.8., page 4). Regularly, clinical phase I studies enroll about 10 to 20 patients. The sample size of 10 patients was chosen due to feasibility reasons. All patients had to be CMV IgG negative and had to be HLA-A*02 positive..

“Patients were enrolled in a two-step 5+5 study design to ensure patient safety as appropriate for a clinical phase I study. The first five patients had to complete all four vaccinations as well as the “end of study visit” 14 days after the last vaccination. Solely one patient per day was allowed to receive the first vaccination within the first five patients. If more than one patient had developed toxicity signs above grade 2, the study would have been stopped. If the true rate of toxicity (> grade 2) is 0.50, then the probability that at least two patients out of five suffer from this event and therefore the early termination of the study is about 97%. The probability to find an event of at least 2 out of ten patients is 99%, when the true rate is 0.5. On the other hand, when the true rate of toxicity is 0.1, the probability to recommend the vaccine is about 93% in the second stage (5-10 patients). This 5+5 design provided the necessary statistical quality for a phase I study.”

  1. Section 2.7, please provide details of what statistical tools were used? e.g. ANOVA etc.

Answer:

Since this are only 10 patients descriptive statistics was used to present data. ANOVA and more statistically analyses would have been not really appropriate.

  1. Section 3.1, a representative data for the release criteria should be shown (it could be provided as supplementary data).

Answer:

We added a new Supplementary Table 2 with representative data for the release of the vaccines.

  1. Line 162. Brief description of method used to test vaccine sterility should be provided in Materials and Methods section.

Answer:

The membrane-bound method was used after validation for bacteria and fungi as required per Ph. Eur. 2.6.1 . in accordance with our Theranostics paper (Reference 19).

  1. Line 164. Under what temperature vaccine was transferred? What was the prescribed range?

Answer:

The temperature was between 4 and 10°C.

  1. Line 167, it is not clear if patients were under constant observation (hospitalized) or stay home?

Answer:

All patients were treated on an out-patient basis during vaccination.

  1. Data in table 3 could be provided per patient, so that correlation with existing disease status could be established.

Answer:

The table is adapted according to the suggestion, devided in two parts a. adverse events associated to vaccination, b. adverse events not associated with vaccination (refer to Table 3 revised).

  1. Section 3.3, second paragraph is not clear. If immune response was detected or not?

Answer:

The sentence was rephrased for better understanding: „In 5 of 10 patients any immune response was detected by an increase in IFNg production in the T-Track™ assay and/or an increase of CMV-specific CD8+ T cells were observed (Table 4, exemplary patient #03 in Figure 1).“

  1. Table 4. Data for SFC should be presented as SFC/ million lymphocytes.

Answer:

We prefer to give the SFC values as per 200.000 cells as per protocol of this commercial assay. The basis of this decision is as follows: Of course, one might multiply the results by 5 to generate values per million lymphocytes. However, this would be a mathematical operation, an extrapolation.  But this would not reflect the reality which is assessed by a lower number of cells. It is a little bit like a magnification by a stronger ocular lens of a light microscope where you have a real limitation by the numeric aperture of your objective.

  1. Please clearly explain the data shown. In the 2nd column, Is this SFC at T0 (time point before vaccination) vs T5 (time point after transplantation or vaccination)?

Answer:

Yes, see Q4 in review 1.

  1. Details of qPCR for CMV should be provided.

Answer:

CMV quantitative PCR (Method section 2.5.) is included in the revised version:

DNA was extracted from 200μl EDTA blood samples and purified using the QIAamp blood kit (QIAGEN, Hilden, Germany) according to the manufacturer’s instructions. A TaqMan real-time PCR assay was performed targeting the UL 86 region in the CMV genome. For quantitative analysis of CMV DNA, 5μL of extracted nucleic acids were amplified with forward primer CMV1 (5'-CAG CCT ACC CGT ACC TTT CCA-3') and reverse primer CMV2 (5'-GCG TTT AAT GTC GTC GCT CAA-3') and detected with the probe 5'-FAM-TTC TAC TCA AAC CCC ACC ATC TGC GC-TAMRA-3'. Additionally, a CMV DNA quantification standard was used threefold in all assays in order to allow quantification of the amplified CMV DNA from patient samples. Quantified CMV DNA was expressed as copies/mL. PCR was performed in a reaction volume of 20μL with a ready-to-use master mix (Roche Diagnostics, Mannheim, Germany) containing Taq DNA polymerase and dNTPs. Amplification and detection were performed on a LightCycler 480 instrument (Roche Diagnostics, Mannheim, Germany) with a thermocycling profile at 95°C for 5 min followed by 50 cycles of 95°C for 5s and 60°C for 20s.

  1. In column 5, "Yes" summarizes results from FACS or ELISPOT or both?

Answer:

See Q1 in review 1.

  1. For Fig 1, please provide a gating strategy. Also, explain why CD8 T cell frequency is so high, and CD8 negative is so low? Generally CD8- population is higher than CD8+.

Answer:

See Q5 in review 1.

  1. Indicate, on the basis of flow results, which patient was responder?

Answer:

See Q5 in review 1.

  1. Since only 5 patients were responders, it is not clear how CMV was controlled in other patients?

Answer:

All patients had a careful follow-up with pre-emptive controls of CMV-PCR every second week after transplantation as written in section 2.1 page 2 last paragraph: „All patients had a pre-emptive CMV therapy after transplantation.“ However, all these patients reactivated and needed treatment against CMV infection. Treatment with ganciclovir intravenously was successful in all patients.

  1. How can the rate of responders be enhanced? Future strategy should be discussed

Answer:

To improve the efficacy of our CMV vaccine, one could add other MHC I and II epitope peptides as a cocktail. Moreover vaccination can be combined with the administration of cytokines or in concert with adoptive transfer of CMV-specific T cells (Schmitt A et al., Transfusion 2011; Neuenhahn et al. Leukemia 2017).

Reviewer 3 Report

The authors reported the result from a Phase I clinical study, where CMV peptide-based vaccines were used to prevent post-transplant CMV infection. The study design was reasonable and the results were carefully analyzed. The article was well organized and well presented. There are some minor points that I would like to point out. 

  1. Standard deviations should be included in the reported values (if applicable)
  2. What was the reasoning for ruling out fatigue as an associated side effect?
  3. Since the heterogeneity among patients' immune responses are inevitable and sometimes unpredictable, would it be better to add pre-vaccination immune tests to further narrow the inclusion criteria?
  4. There are a few typos in the manuscript. Please check carefully.

Author Response

Reviewer 3

Comments and Suggestions for Authors

The authors reported the result from a Phase I clinical study, where CMV peptide-based vaccines were used to prevent post-transplant CMV infection. The study design was reasonable and the results were carefully analyzed. The article was well organized and well presented. There are some minor points that I would like to point out.

Thank you very much for your assessment. We greatly appreciate your time and effort for our manuscript.

  1. Standard deviations should be included in the reported values (if applicable)

Answer:

Standard deviations are provided whereever applicable in the revised version.

  1. What was the reasoning for ruling out fatigue as an associated side effect?

Answer:

Fatigue was documented if it was reported by the patients. Fatigue is added to the vaccination associated side effects in the revised version of the manuscript.

  1. Since the heterogeneity among patients' immune responses are inevitable and sometimes unpredictable, would it be better to add pre-vaccination immune tests to further narrow the inclusion criteria?

Answer:

This item is added in the last part of the discussion section: “Since immune response among the patients´ is heterogenous and sometimes unpredictable, a pre-vaccination immune test as the CD4 T cell count might be helpful to detect patients with potential response to vaccination.”

  1. There are a few typos in the manuscript. Please check carefully.

Answer:

We corrected the entire manuscript carefully for spelling and grammar.